# Effect of Loading Angles and Implant Lengths on the Static and Fatigue Fractures of Dental Implants

**DOI:** 10.3390/ma14195542

**Published:** 2021-09-24

**Authors:** Fei Sun, Li-Tao Lv, Wei Cheng, Jia-Le Zhang, De-Chun Ba, Gui-Qiu Song, Zeng Lin

**Affiliations:** Key Laboratory of Implant Device and Interface Science of Liaoning Province, School of Mechanical Engineering and Automation, Northeastern University, Shenyang 110819, China; 1810132@stu.neu.edu.cn (F.S.); 2070303@stu.neu.edu.cn (L.-T.L.); 1970085@stu.neu.edu.cn (W.C.); 1970285@stu.neu.edu.cn (J.-L.Z.); dchba@mail.neu.edu.cn (D.-C.B.); gqsong@mail.neu.edu.cn (G.-Q.S.)

**Keywords:** dental implant, loading angles, implant length, static failure, fatigue fracture

## Abstract

Mechanical properties play a key role in the failure of dental implants. Dental implants require fatigue life testing before clinical application, but this process takes a lot of time. This study investigated the effect of various loading angles and implant lengths on the static fracture and fatigue life of dental implants. Implants with lengths of 9 mm and 11 mm were prepared. Static fracture tests and dynamic fatigue life tests were performed under three loading angles (30°, 40°, and 50°), and the level arm and bending moment were measured. After that, the fracture morphology and fracture mode of the implant were observed. The results showed that 9 mm length implants have a higher static failure load and can withstand greater bending moments, while 11 mm length implants have a longer fatigue life. In addition, as the loading angle increases, the static strength and bending moment decrease linearly, and the fatigue life shows an exponential decrease at a rate of three times. Increasing the loading angle reduces the time of the implant fatigue test, which may be an effective method to improve the efficiency of the experiment.

## 1. Introduction

The mechanical properties of dental implant have always been a popular research topic. The mechanical complications of dental implants include fracture or loosening of the abutment and abutment screw and implant fracture. Studies have shown that the long-term incidence of mechanical complications of implants ranges from 21% to 31% and is greatly affected by the structural design of the implant [1,2,3].

The static load test is usually used to determine the fracture strength or bearing capacity of implant components, while the cyclic fatigue load test aims to study its mechanical durability [4]. Under static load and cyclic load conditions, the mechanical behaviour of the implant system is different [5]. The stress of the specimen under cyclic loading is much lower than that under static loading. When the stress level is lower than the fatigue limit, no fracture will occur. The fatigue limit refers to the maximum life of a component without damage under certain stress cycles. From a clinical point of view, fatigue can lead to serious consequences. In order to make dental implants have good biocompatibility and mechanical properties in clinical practice, the most commonly used material for abutments and abutment screws is Ti-6Al-4V, and the most often used material for implants is grade 4 pure titanium [6]. In addition to the characteristics of the material itself, the load fatigue performance and failure location are system-specific and are directly related to the implant system design (the size of the crown, abutment, implant, etc.) and the load type (the direction and angle of loading).

In terms of implant system design, different structures have different mechanical properties. Compared with other implant-abutment connections, tapered internal connection implants have higher resistance to torque loss, higher fatigue resistance, higher bending resistance, and lower abutment screw stress [7]. In addition, increasing the abutment taper can significantly increase the fracture resistance of the implant. This is due to the increase in the taper angle, which results in an increase in the wall thickness of the implant in the connecting part [8,9]. Finite element analysis shows that the increase in implant length and diameter reduces the stress changes in the surrounding bone and the stress distribution in the implant, but the structure design of the implant has a more obvious effect than the length of the implant [10,11,12,13]. Research on the crown height shows that increasing the crown height will increase the stress concentration of the implant components and reduce the connection stability [14,15].

In terms of load types, mechanical tests based on the ISO 14801 standard are useful for comparing implants of different designs, sizes, and assembled states [16]. These tests show the structural failure mode of the implanted system. However, the limitation of the ISO 14801 standard lies in the limitation of the load angle and load frequency, as well as the clamping position. Therefore, the modified form of ISO 14801 is often used in the study of the mechanical properties of dental implants. For example, in a liquid environment, the fatigue life and failure mode of the implant system are not significantly different between 2 Hz and 15 Hz [17]. However, comparison tests at 2 Hz and 30 Hz in air found that despite having the same fracture model, the implant system had a lower fatigue life under the 2 Hz conditions [18]. Comparing the mechanical properties of the three loading conditions (bending moment, torsional moment, and axial loading) of the taper connection, it is found that due to the bending moment, the taper connection produces obvious fretting wear, which leads to the loss of torsion resistance, but increasing the axial load can significantly improve its torsion resistance [19]. In addition, another study showed that compared with single directional loading, multidirectional loading conditions produce a lower fatigue life [20].

Although the effects of implant system design and load types on the mechanical properties have been extensively studied, there is a lack of research on the effects of different loading angles on the mechanical properties of implants [21,22]. Moreover, before clinical application of dental implants, it will take a lot of time to complete the fatigue life test according to the ISO14801 standard. This paper studies the influence of implants of different lengths on the static strength and fatigue strength under different loading angles. The purpose of this study is to determine the relationship between the loading angle and fatigue life. Furthermore, when performing standard certification for dental implants, the fatigue test time can be shortened by increasing the loading angle. The research hypothesis was that the no difference would be found between implant lengths and loading angles on the static strength and fatigue strength.

## 2. Materials and Methods

The dental implants (Ø 3.8 × L 9 mm and Ø 3.8 × L 11 mm) were made of commercially pure titanium grade 4 (cpTi grade 4) and the straight abutments (Ø 4.5 × Gh 4.5 × h 6.0 mm) and abutment screws (Ø 1.5 × L 13.5 mm) were made of titanium alloy (Ti-6Al-4V). The structure of the test sample is shown in Figure 1. All the test samples were manufactured by WEGO (Jericom Biomaterials Co., Ltd., Weihai, China). A total of 36 samples were prepared and divided into two groups according to the implant length: the 9 mm implant group (SG: 9 mm implant system group) and the 11 mm implant group (LG: 11 mm implant system group).

The test model was developed according to standard ISO14801:2016 (Dynamic Fatigue Test of Dental Intraosseous Dental Implants, Figure 2A), which specifies that the loading angle between the applied force and the axis of the implant is 30° and the distance between the implant and the platform is 3 ± 0.5 mm to simulate clinical bone loss. In order to study the influence of the loading angle on the mechanical properties, the loading angles of the implants of 30°, 40°, and 50° were selected (Figure 2B–D). The lever arm (LA) was calculated using the formula LA = l × sin θ, where l is the distance between the abutment point of the load and the clamp-implant junction (11 mm) and θ is the loading angle. The bending moment (M) was calculated using the formula M = F × LA (ISO 14801: 2016), with F being the load (N) and LA being the lever arm (in cm). The implant system consisted of implants, abutments, abutment screws, and simplified crowns, which were fixated with metal clamps. The connection was tightened to 30 N cm.

The static fracture test module and fatigue test module were used in the dynamic fatigue testing machine (CARE M-3000, Tianjin, China). The static fracture test was performed at a speed of 0.05 mm/s until the implant system broke. Next, the maximum load was recorded and defined as the failure load. The static fracture test was conducted three times in each group. After the static fracture test, the dynamic fatigue test was performed used a dynamic sinusoidal load with a load of 30 to 300 N, a frequency of 15 Hz. The fatigue life was determined as the number of cycles until failure. The dynamic fatigue test was conducted three times in each group. A scanning electron microscope (SEM, ULTRA 55, Carl Zeiss, Germany) was used to examine the surface morphology of the fatigue fracture. After observing the fracture morphology, the fracture modes were compared in the SG and LG with different loading angles.

The 3D models were created by SolidWorks 2016 software (Dassault Systèmes SolidWorks Corp., Concord, MA, USA) according to the design and size of the implant samples. As shown in Figure 3, implants of different lengths were inserted into the metal block at 30°, 40°, and 50° off-axis. The load direction was identical to the axis of the fixture.

Three-dimensional models were incorporated into the FEA software (Ansys Workbench 18.0, Swanson Analysis Inc., Houston, PA, USA). cpTi grade 4 material was assigned to the implant, Ti-6Al-4V material was assigned to the abutment and abutment screws, and stainless-steel material was assigned to the crown and fixture. The von Mises stresses were used to evaluate the stress of the implant system. The convergence check was performed by optimising the number of networks until the value change was less than 5%. The number of elements and the number of nodes are shown in Table 1. According to the FEA model, (1) all materials were homogeneous, isotropic, and elastic; (2) the physical properties are shown in Table 2; (3) “perfect bonding” was defined for different component interfaces, and a “fixed” constraint was applied to the metal fixture; and (4) the parts were free of defects. 

According to the literature [24], a preload of 260 N was added to the abutment screw. Furthermore, an external load of 200 N was applied to the simplified crown surface. The results of SG and LG were analysed uniformly, and the changes in the law of loading angle on failure load, bending moment, and fatigue life were studied. SPSS software was used to conduct statistical analysis, and data were analysed with Fisher’s PLSD and ANOVA. *p* < 0.05 was defined as statistically significant.

## 3. Results

After the static failure test, all the samples showed irreparable plastic deformation at the connection between the metal fixture and the implant. Table 3 lists the lever arm, failure load, and bending moment changes for three loading angles. In the correlation analysis of the failure load and bending moment values of the three angles, it was found that the SG and LG at loading angles of 30° and 40° showed statistically significant differences (*p* < 0.05 between the groups at the same loading angle), while no significant differences were detected between the SG and LG at a loading angle of 50° (*p* > 0.05). As the external load increased, the deformation of the implant system increased accordingly. Failure occurred when the deformation exceeded the material limit. The loading process of different specimens is shown in Figure 4, and the peak value represents the maximum load before failure. The SG had a greater failure load than the LG, and as the loading angle increased, the failure load value decreased. From the calculation results of the bending moment, the SG also had a larger bending moment, and the bending moment values of the SG and LG gradually decreased as the loading angle increased. 

After the fatigue life test, all samples showed irreparable brittle fractures at the connection between the metal fixture and the implant. The fatigue life test results are shown in Figure 5. As the loading angle increases, the number of cycles gradually decreases. When the loading angle is 30°, there is a large difference in the number of cycles. When the loading angle increases to 50°, the difference in the number of cycles decreases. The length of the implant affects the fatigue life. The LG experiences more cycles than the SG. There is no significant difference between the groups at the same loading angle (*p* > 0.05). 

Figure 6 and Figure 7 are the fatigue cross-sections of the SG and LG. The fracture positions of the two implant systems are the same, including implant fracture and abutment screw fracture. The fracture surface of the implant is the contact area between the neck and the fixture. Abutment screw fracture occurred at the root of the first level thread, and the abutment was undamaged. Figure 6 shows the fracture morphology of the SG under different loading angles. The yellow arrow is the fracture direction. The initiation and propagation of cracks are observed on the surface of the implant fracture (Figure 6A1–C1). The direction of fatigue striations is perpendicular to the direction of fracture. Along the fracture direction, dimples are observed on the fracture surface of the abutment screw (Figure 6A2–C2). In addition, Figure 7 shows the fracture morphology of the LG at different loading angles, showing similar results to that of the SG. 

Under an external force of 200 N, the equivalent stress distributions of the SG and LG components with loading angles 30° are shown in Figure 8. The maximum von Mises stress positions are concentrated: the maximum stress of the implant is located in the implant-fixture connection area, the maximum stress of the abutment is located at the taper fit of the implant-abutment, and the maximum stress of the abutment screw is located at the first of the implant-abutment screw thread connection. Figure 9 shows the stress values of the SG and LG components under three loading angles. As the angle increases, the stress value of each component increases. The implants of the SG and LG are the most stressed components, indicating that the implants are the most easily damaged components. Under the same loading angle, the implant stress value of the SG is smaller than that of the LG.

Figure 10 was the box plot diagram of the static failure load, bending moment and fatigue life under three loading angles. Compared with the loading at 30°, the static load failure value is reduced to 67% and 51% at 40° and 50°, respectively (Figure 10A). The failure load is significantly different (*p* < 0.05 among groups) among the three loading angles. As shown in Figure 10B, the increase in the loading angle changes the bending moment value, but it is not obvious compared with the static load failure. Compared with the scenario of loading at 30°, the bending moment values are at 40° and 50°, which drop to 87% and 78%, respectively. There is no significant difference in the bending moment value between the loading angles of 40° and 50°, but there are significant differences (*p* < 0.05 among groups) between the other two groups of loading angles. As shown in Figure 10C, the increase in the loading angle significantly reduces the fatigue life. Compared with the loading at 30°, the fatigue life values are reduced to 33% and 11% at 40° and 50°, respectively. There are significant differences (*p* < 0.05 among groups) in the number of cycles among the three loading angles.

## 4. Discussion

The research hypothesis that the no difference would be found between implant lengths and loading angles on the static strength and fatigue strength was rejected. The results show that the implant lengths and loading angles changed static strength and fatigue strength of the dental implants.

Experimental testing is a reliable and useful method for determining the mechanical properties for implant research and development purposes. It is repeatable and can compare and analyse the mechanical response of implants of different designs under the same load conditions [8]. When investigating the effects of size, material and external loads, the same implant system is usually chosen for comparison purposes [25,26,27]. To analyse the impact of the implant length on the damage strength, the same abutment and abutment screw were selected throughout the analysis.

In the static failure test, the samples showed irreparable plastic deformation. As shown in Figure 4, it can be found that when the displacement was less than 1 mm, the implant system was mainly in the elastic deformation stage. At this stage, the load–displacement curve was in a linear relationship, which conforms to the Hooke’s law of materials. When the displacement was greater than 1 mm, it was the plastic deformation stage of the implant system. At this stage, the load–displacement curve was no longer in a linear relationship, and the change trend is affected by the implant length. The results showed the SG had a greater failure load than the LG, and the structure of the implant system had a significant impact on the failure load [28]. The static load test was performed on the implant system to evaluate its mechanical properties, and the results showed that it has sufficient strength to withstand the oral bite force [29,30]. The damage is caused by the bending moment, which causes permanent deformation of the implant components. Considering clinical safety, a minimum bending moment of 200 N cm is sufficient [31]. When this requirement is applied to the results, all the test results meet the requirements.

In the fatigue life test, the samples showed irreparable brittle fractures. Under static failure test and dynamic fatigue test, the implant system exhibited different mechanical behaviours. Static load shows the fracture strength or bearing capacity of the implant, while fatigue life reflects its mechanical durability [5]. As shown in Figure 5, the number of cycles gradually decreases as the loading angle increases [22], and LG has a longer fatigue life than SG. The fatigue test obtained the opposite result for static load failure. In the taper internal connection system, the implant-abutment mating is produced by surface friction. Due to the principle of Morse taper, the high friction force connects the two surfaces perfectly. This phenomenon is called cold welding. Because the surface is rough at the microscopic level, the contact pressure causes penetration and fusion between the surface asperities. Cold welding makes the system almost a whole, and the stress distribution is uniform throughout the system [32]. Therefore, the result is similar to the mechanical properties of most structural materials, that is, strength and toughness are usually mutually exclusive [33].

As shown in Figure 6 and Figure 7, this failure mode is similar to the failure mode of the ITI implant designed with an 8° TIS connection [34]. The Morse taper connection between the implant and the abutment will increase the friction between the interfaces and produce cold welding to protect the abutment screw from fracture [35]. The fracture surface shows the fracture process of the entire implant system, and typical fatigue characteristics can be observed according to the surface topography. Although the fatigue life of the two groups of implant systems is different, there is no change in surface topography.

The mechanical testing shows when and where the system will be damaged, but the FEA method provides insight into the inherent mechanics of a given technical system. It can describe the internal stress situation and show the location of weak points in the system [32]. FEA has been utilised in dental implant research to investigate the distribution of stress in key components and assess the biomechanical performance of different loading conditions [36,37]. As shown in Figure 9, the implants are the most stressed components. According to the difference of the purity of titanium and the content of other elements, it can be divided into 5 grades. Grades 1 to 4 are non-alloyed cpTi and grade 5 is alloyed Ti-6Al-4V. The higher the grade, the higher the strength of the titanium material [38,39]. Because the implants are made of cpTi grade 4, and the abutment and abutment screw are made of Ti-6Al-4V, the yield strength and ultimate tensile strength of cpTi grade 4 are less than Ti-6Al-4V [40,41], which also shows that the implants are easier to destroy. This result is consistent with the results obtained from the static experiment, and the fracture position is also verified, which is the implant neck area. Under the same load, the implant stress value of the SG is smaller than that of the LG. This shows that the SG can bear a greater load, and the load value in the static failure test is greater than that of the LG. The abutment stress value of the SG is greater than that of the LG, indicating that the internal stress transmission of SG is not as good as that of LG [13]. The SG and LG showed similar abutment screw stress values, indicating that the length of the implant has little effect on the stress value of abutment screw, and the increase in the loading angle has a small effect on the stress value of abutment screw [42,43]. The stress value of the abutment screw is mainly affected by the preload force.

Dental implant products need to be tested for fatigue life before clinical use, which meets the industry standard ISO 14801, and the fatigue life reaches 5 million times under 30 degree loading angle. However, due to the requirements of the number of cycles, a single sample test sometimes takes many days to complete, and a set of tests may even take dozens of days. Therefore, we propose to shorten the fatigue test time by increasing the loading angle. In order to study the influence of loading angle on failure load, bending moment and fatigue life, the results obtained by SG and LG were summarised and analysed. As shown in Figure 10, it can be seen that as the loading angle increases, the static failure load decreases significantly, and the overall change is linear. Although the overall change of the bending moment is also linear with the increase in the loading angle, the decrease in the bending moment is relatively slow. Bagegni studied the failure bending moment values of the internal hexagonal connection and the internal cone connection, which are 518 N cm and 534 N cm, respectively [15]. This is closer to the value obtained by our experiment (>480 N cm). This shows that the implant system has certain requirements for the bending moment value to achieve static load failure. Finally, the increase in the loading angle significantly reduces the fatigue life. It can be seen that the decline is relatively rapid, and the overall decline is exponential. When the loading angle increases by 10 degrees, the fatigue life is shortened to one-third. In other words, for this implant system, the fatigue life of loading angle 50 degrees is equivalent to the fatigue life of loading angle 30 degrees, but only one-ninth of the test time is used. Therefore, we propose a method based on standard ISO14801 to shorten fatigue test time and increase the fatigue test efficiency by increasing the loading angle. Of course, the general applicability of this method needs to be systematically studied in the future. This research also has some limitations, such as whether there is the same proportional relationship with other structural implants under high-cycle fatigue loading. In the FEA, the properties of the materials, Ti and Ti-6Al-4V were acquired from other scientific research and modeled with isotropic, linear, homogeneous, and elastic properties.

## 5. Conclusions

Based on the findings of this in vitro and FEA study, the following conclusions were drawn: 9 mm implants have a higher static failure load, while 11 mm implants have a longer fatigue life. Increasing the loading angles reduces the static strength, bending moment and fatigue life of the implant. There is an exponential relationship decreasing at a rate of three times between the loading angle and fatigue life. Based on standard ISO14801, increasing the loading angles may be an effective way to shorten test time and improve the efficiency of the experiment. 

## Figures and Tables

**Figure 1 materials-14-05542-f001:**
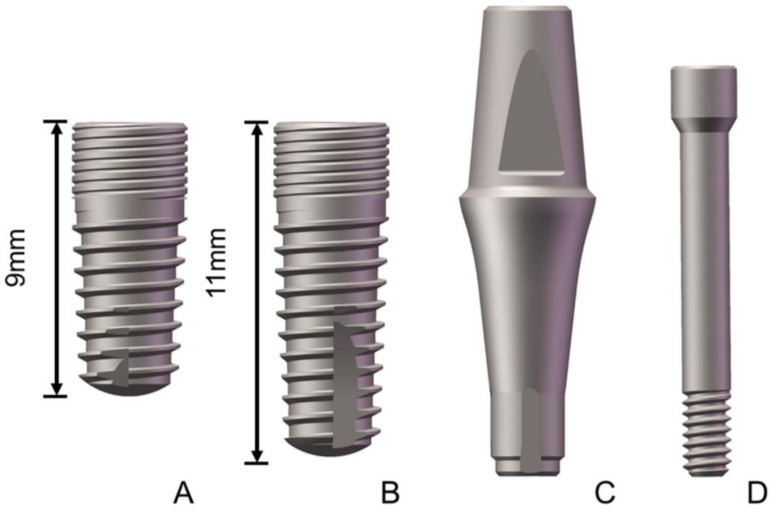
Structure diagram of the tested samples ((**A**): 9 mm implant; (**B**): 11 mm implant; (**C**): abutment; and (**D**): abutment screw).

**Figure 2 materials-14-05542-f002:**
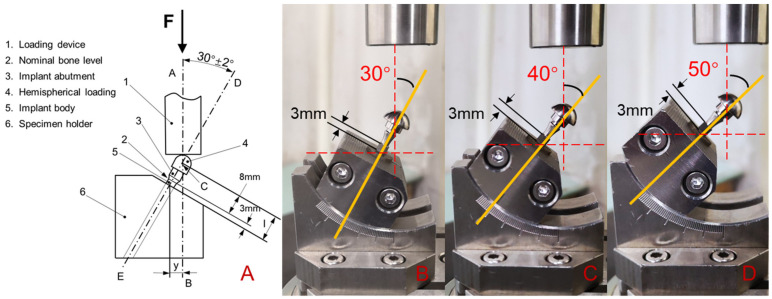
Dynamic loading apparatus for the fatigue test ((**A**): schematic diagram of the cyclic loading device; (**B**): loading angle 30°; (**C**): loading angle 40°; and (**D**): loading angle 50°).

**Figure 3 materials-14-05542-f003:**
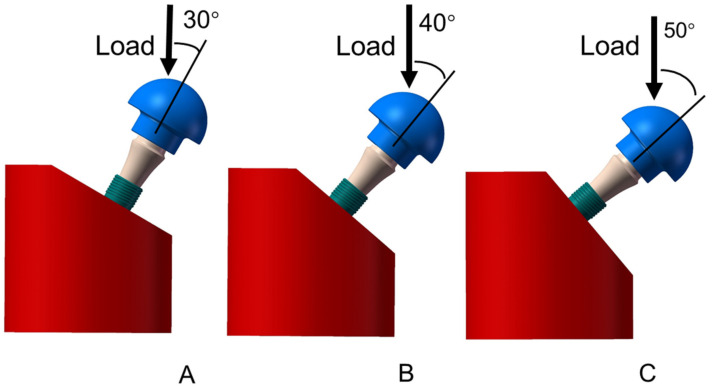
Finite element model structure of the SG and LG ((**A**): loading angle 30°; (**B**): loading angle 40°; (**C**): loading angle 50°)**.**

**Figure 4 materials-14-05542-f004:**
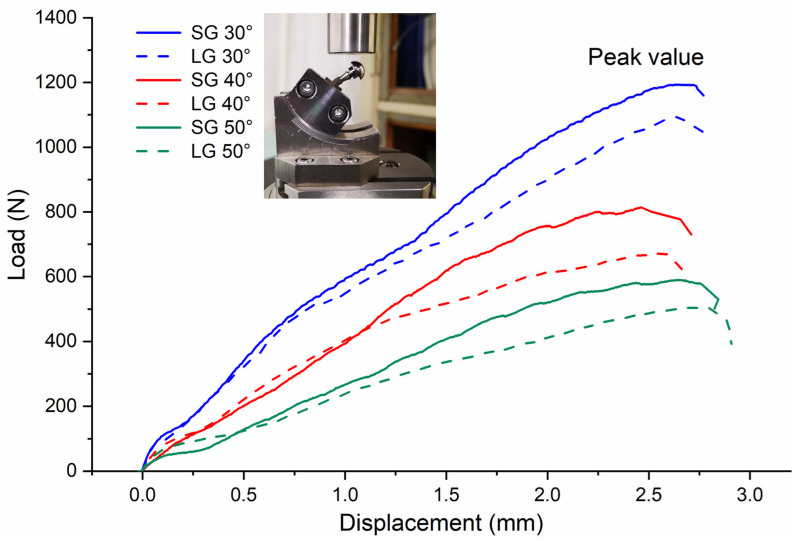
Load–displacement curve from the static load failure test.

**Figure 5 materials-14-05542-f005:**
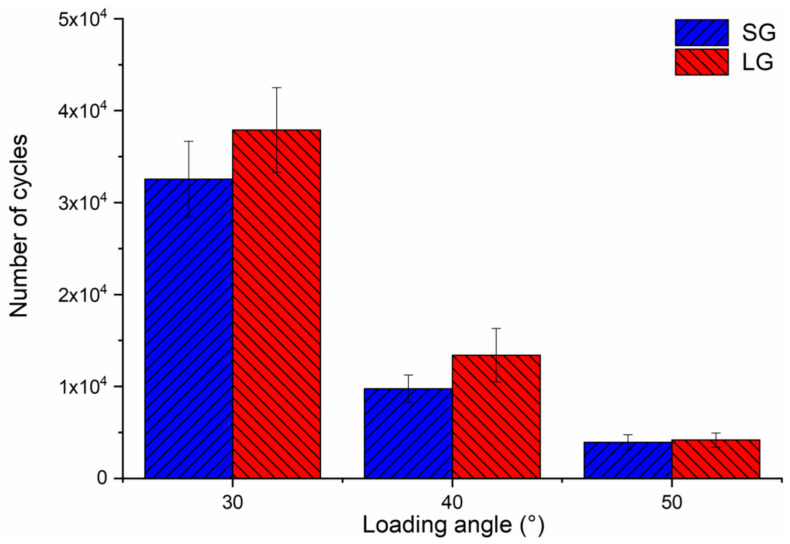
Diagram of the fatigue life under three loading angles.

**Figure 6 materials-14-05542-f006:**
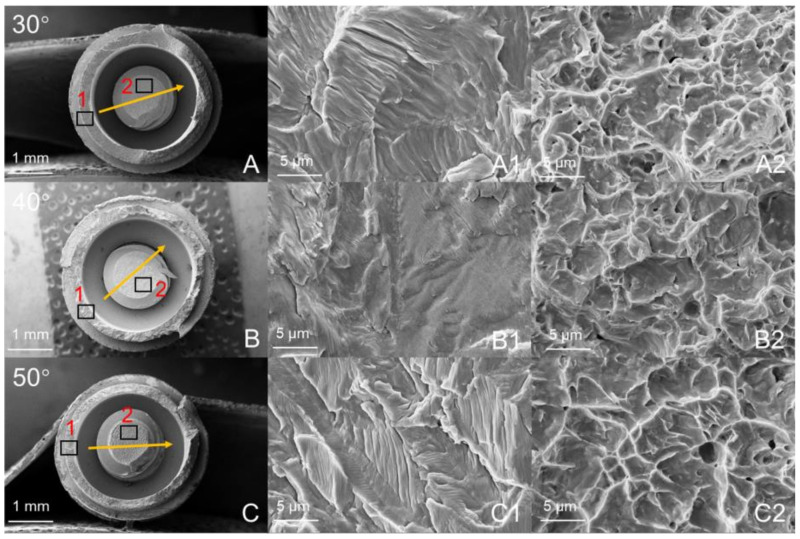
SEM images of the fracture surfaces of the SG: (**A**–**C**) Integral morphology; (**A1**–**C1**) area shown by “1”; (**A2**–**C2**) area shown by “2”. The arrow indicates the direction in which the fracture extended.

**Figure 7 materials-14-05542-f007:**
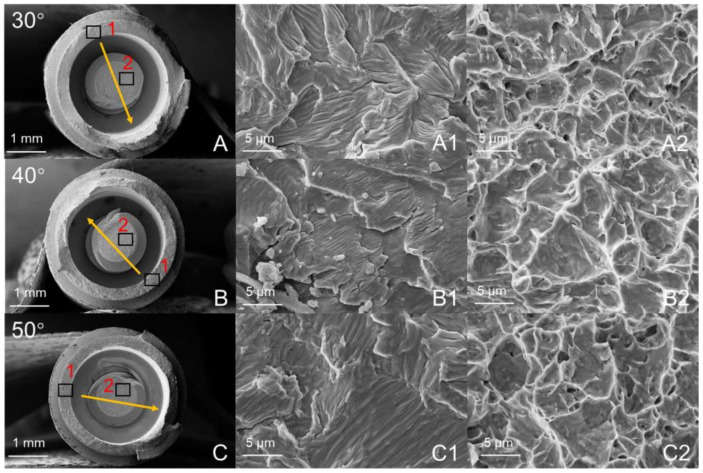
SEM images of the fracture surfaces in the LG: (**A**–**C**) Integral morphology; (**A1**–**C1**) area shown by “1”; (**A2**–**C2**) area shown by “2”. The arrow indicates the direction in which the fracture extended.

**Figure 8 materials-14-05542-f008:**
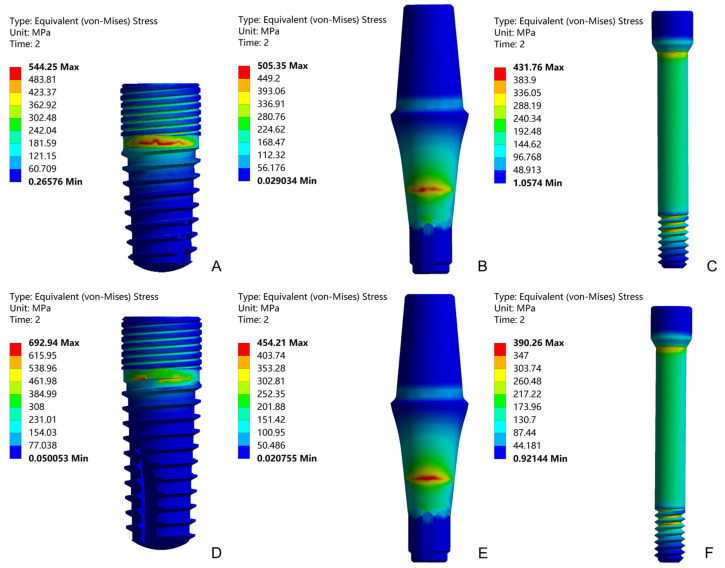
Equivalent stress distributions of the implant, abutment, and abutment screw with loading angles 30°: SG (**A**–**C**) and LG (**D**–**F**).

**Figure 9 materials-14-05542-f009:**
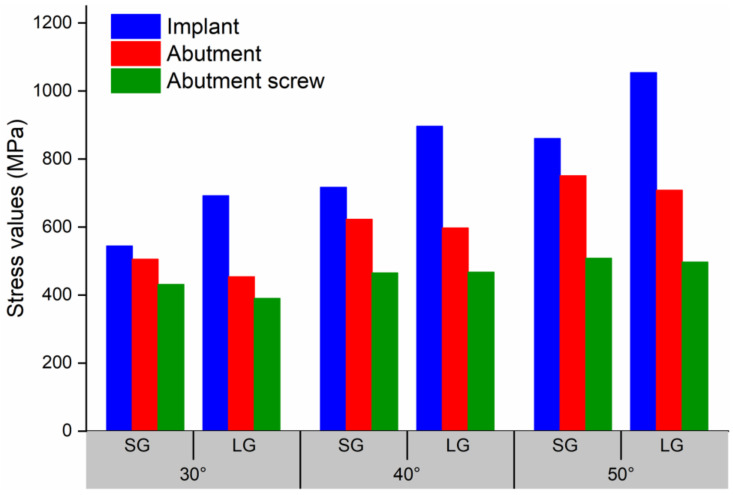
The Maximum von Mises stress values of the SG and LG under three loading angles.

**Figure 10 materials-14-05542-f010:**
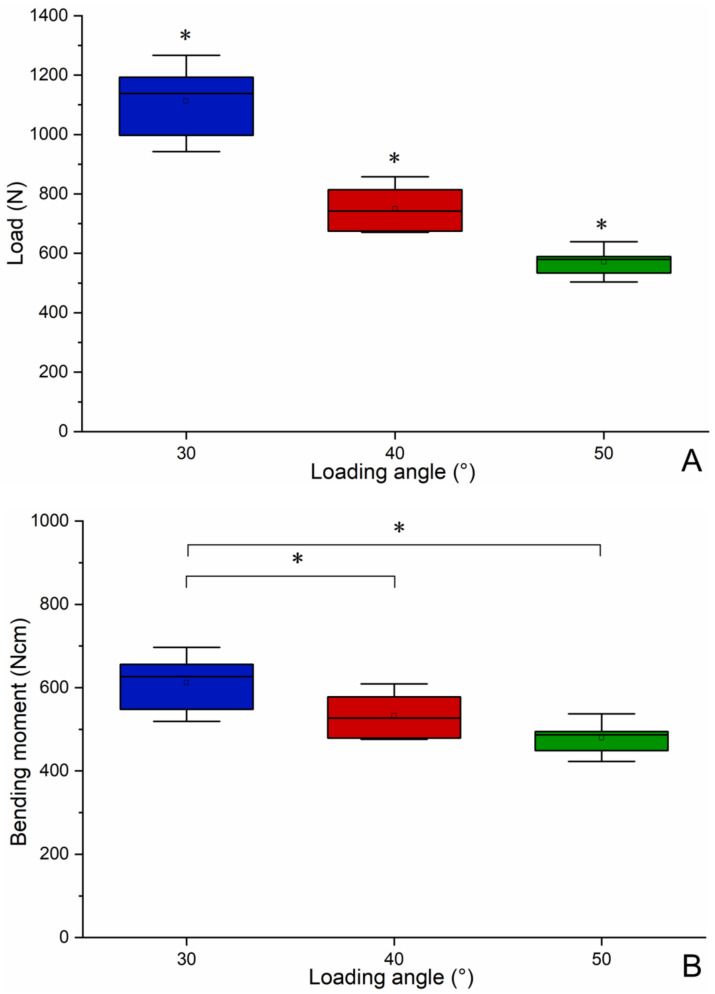
Boxplot diagram of the static failure load (**A**), bending moment (**B**) and fatigue life (**C**) under three loading angles (* significant difference between groups).

**Table 1 materials-14-05542-t001:** Number of elements and nodes.

	SG 30°	SG 40°	SG 50°	LG 30°	LG 40°	LG 50°
Number of elements	125,471	126,105	125,856	131,520	131,999	131,765
Number of nodes	193,632	194,547	194,148	203,369	204,032	203,655

**Table 2 materials-14-05542-t002:** Physical properties of the materials for FEA.

Material	Elastic Modulus (GPa)	Poisson’s Ratio	Source
cpTi grade 4	102	0.3	[23]
Ti-6Al-4V	110	0.3	[23]

**Table 3 materials-14-05542-t003:** Lever arm, failure load and bending moment changes for three loading angles (The same letters (a, b, A and B) indicate that *p* < 0.05 by comparing the values).

Group	Loading Angle	Lever Arm (cm)	Failure Load (N)	Bending Moment (N cm)
SG	30°	0.55	1214.3 ± 37.5 ^a^	667.9 ± 20.6 ^A^
	40°	0.71	811 ± 39.7 ^b^	575.8 ± 28.2 ^B^
	50°	0.84	601 ± 27.5	504.8 ± 23.1
LG	30°	0.55	1011.3 ± 62.5 ^a^	556.2 ± 34.4 ^A^
	40°	0.71	690.0 ± 24.1 ^b^	489.9 ± 17.1 ^B^
	50°	0.84	540.7 ± 33.0	454.1 ± 27.7

## Data Availability

Data are available on request to the corresponding author.

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
