# Peer review of "Effect of Loading Angles and Implant Lengths on the Static and Fatigue Fractures of Dental Implants"

_materials, 2021, doi:10.3390/ma14195542_

Round 1

Reviewer 1 Report

The authors are commended for their well written and described study which will be of significance to users and developers of dental implants.

All tests appear to have been done in air and as such how well do the results relate to clinical reality in the oral cavity?

There is one glaring issue with the language in the Introduction that is repeated at the start of the Discussion.  Namely P2 L78   .... be found between implant lengths and ...  (that is add "between" found and implant). Similarly in the Discussion P12 L225.  

Reviewer 2 Report

The authors presented an interesting topic related to the static and fatigue fractures of dental implants with different lengths and loading angles. The topic is very interesting, however the manuscript requires a few corrections and additions.

In the introduction, the authors should describe in detail the titanium alloys - most often used in dental implantology. Information on the percentage of individual elements content in a given alloy should be included, e.g . Prosthesis 2020, 2, 100–116; doi:10.3390/prosthesis2020011; RSC Adv., 2018, 8, 15533–15546 | 15533; Materials (Basel). 2014 Dec; 7(12): 8168–8188.

Please include in the discussion - how the composition of the alloy can influence the mechanical properties of implants.

Reviewer 3 Report

Dear authors, thanks to provide this research. Even if I think this research is well presented and written, I do not agree with the methodology. I studied a lot the problem of implants fracture. Implant diameter and overloading are two of the major risk factors, I agree. Nevertheless, I noted that you simulated a bone level implants, 3 mm above the bony crest, with a 3.8 mm diameter implant, with conical connection, that is not indicated in the area with high occlusal forces, and with a maximum loading angle of 50° that is quite not real in daily practice for a single tooth. However, is not prosthetically correct. So, I think this study do not offer any usable informations for further clinical report, and of course, for readers. On the contrary, may create confusion. Longer narrow implants, placed above the bone crest, with 30° of loading angle are still at risk of fracture, in the molar area.

In addition, please check my step-by-step considerations.

Study design must to be reported in the title.

I would like to read a structured abstract, but please check the authors guidelines.

Abstract, line 10. What does "dental implant systems" mean? I prefer "dental implants" without systems, or "implant-supported prostheses".

Introduction

Lines 29 to 33, please provide updated references and limit to systematic reviews.

Line 45. I prefer "connections" instead of "connection methods,"

Lines 47,48. Regarding the sentence: "In addition, increasing the taper can significantly increase the fracture resistance of the implant." Please, specify that is the "abutment taper angle". Otherwise, seems the taper of the implant.

Lines 73-74. "there is a lack of research on the effects of dif-73 ferent loading angles on the mechanical properties of implants." For example, in the literature I found this paper of 2009. "Qian, L., Todo, M., Matsushita, Y., & Koyano, K. (2009). Effects of implant diameter, insertion depth, and loading angle on stress/strain fields in implant/jawbone systems: finite element analysis. The International journal of oral & maxillofacial implants, 24(5), 877–886." Please add some references.

In the manuscript is written "The dental implants (Ø 3.8×L 9 mm and Ø 3.8×L 11 mm) were made of pure titanium (Ti)". I failed to find more information in the manufacturer's web site. Please specify "pure titanium". Commercially Pure Titanium Grade 1 is the softest titanium. Please clarify.

Avoid personal pronoun in the text.

This research also has some limitations, such as whether there is the same proportional relationship with other structural implants under high-cycle fatigue loading. In my personal opinion, there are more limitations.

I do not understand why the simulated implants are 3 mm above the bony crest. Simulated implants are bone level implants, not tissue level implants.

Moreover, only one diameter has been used. It is well known that the diameter is most important that the implant length. Moreover, most implant companies suggested 4.5 mm diameter implants for molar region, or stronger implants, such as titanium alloy implants.

The 9 mm length implants are not short implants. This may create confusion.

As demonstrated in the past, increasing the angle, the resistance of the implants reduce. Nevertheless, the limitations of the present research still remain. In vitro study are very different that clinical report. 50° of loading angle in a single crown with an implant positioned 3 mm above the bony crest is not a real situation, I hope.

Reviewer 4 Report

The manuscript “Comparison of the static and fatigue fractures of dental implants with different loading angles and implant lengths” deals with an actual problem related to static and fatigue fractures od dental implants. It is a competent experimental study with detailed examination of two different types of dental implants.

The introduction in the manuscript presents a clear background of the conducted research. The Authors cite both literature sources in the form of scientific publications as well as in the form of a standard, which indicates a good understanding of the authors in this type of research.

In case of methods and materials section the experimental procedure is given in due details. Additional experimental results are clearly written, while the comparison with the relevant literature is performed.

Conclusions logically summarizes the obtained results.

The following remarks, being taken into account, can improve the manuscript.

  1. Why the Authors decided to perform tests using angles of 30, 40 and 50 degrees. If such values are specified in the standard, please correlate it more precisely in the description of the experiment. If not, please explain.
  2. The designations of statistical differences (a, b, A and B) in table 3 should be described in the description above the table.
  3. The SEM photos in figures 6 and 7, although in good resolution, do not have a clear scale. The scale should be enlarged.
  4. In Figure 4, the Authors present the load-displacement curve from the static load failure test for long and short implants with loading angles 30, 40 and 50. How the authors explain the fact that in the case of implants with the same angle in the initial phase of the force increase, the curves practically overlap on yourself. After a displacement of approx. 1 mm, changes in the curves begin to become visible.
  5. There is no scale bar on figure 8.

Reviewer 5 Report

The authors present a study on the comparison of the static and fatigue fractures of dental implants with different loading angles and implant lengths.

The introduction defines the mechanical properties of dental implants (abutment, abutment screw, and implant fracture), static load, and cycle load, FEA analysis of dental implants, described standard ISO 14801. Finally, the aim of the study and its hypothesis is mentioned. The authors could add to the significance of the presented study for the dental practice.

Material and Methods - In this chapter, the authors defined the choice of the dental implant (material, dimensions), described the test model. It determined the conditions for static and dynamic testing, as well as for SEM analysis and FEA analysis. 
Based on what did the authors choose the 9 and 11 mm length of the dental implant?
On what basis did the authors choose an angular bearing for a dental implant? Is the application of a dental implant at a 50° angle used in dental practice?
What material was used for the part of the test model in which the dental implant was attached? Wouldn't it be more appropriate to choose a material that has similar properties to the human bone into which the dental implant is applied?

Result - Test results are clearly described, processed into tables and graphs. 

Discussion - The Discussion clearly compares the results with the results from other relevant studies.

The presented study is clearly described, the procedures are clearly defined, the results are clearly processed and supplemented with suitable graphs. However, I lack a connection with practice - how important are the results for a dental practice, how can they apply the results in practice. I ask the authors to complete it.

Thank you

Round 2

Reviewer 3 Report

Dear authors thanks to provide a new version of your research. Thanks to fix most of my comments and to provide explanation for the others.

I just suggested to improve the discussion providing limitation of the ISO14801. I agree that ISO14801 could be considered standard for industry, but this research is not aimed to certified some implants. It is aimed to evaluate various loading angles and implant lengths on the static fracture and fatigue life of dental implants. 

You provided to me a good explanation. I would like that you provide a short paragraph in the discussion.

Please, also check this sentence on line 77. 

The purpose of this study is to determine the relationship between the loading angle and fatigue life to shorten the test time. 

It is confused.

Thanks
